# Methyl Parathion Exposure Induces Development Toxicity and Cardiotoxicity in Zebrafish Embryos

**DOI:** 10.3390/toxics11010084

**Published:** 2023-01-15

**Authors:** Tianyi Chen, Haoze Chen, Anli Wang, Weixuan Yao, Zhongshi Xu, Binjie Wang, Jiye Wang, Yuanzhao Wu

**Affiliations:** 1Key Laboratory of Drug Prevention and Control Technology of Zhejiang Province, The Department of Criminal Science and Technology, Zhejiang Police College, Hangzhou 310053, China; 2National Engineering Laboratory of Intelligent Food Technology and Equipment, Zhejiang Key Laboratory for Agro-Food Processing, Fuli Institute of Food Science, College of Biosystems Engineering and Food Science, Zhejiang University, Hangzhou 310058, China

**Keywords:** methyl parathion, cardiotoxicity, oxidative damage, zebrafish, apoptosis

## Abstract

Methyl parathion (MP) has been widely used as an organophosphorus pesticide for food preservation and pest management, resulting in its accumulation in the aquatic environment. However, the early developmental toxicity of MP to non-target species, especially aquatic vertebrates, has not been thoroughly investigated. In this study, zebrafish embryos were treated with 2.5, 5, or 10 mg/L of MP solution until 72 h post-fertilization (hpf). The results showed that MP exposure reduced spontaneous movement, hatching, and survival rates of zebrafish embryos and induced developmental abnormalities such as shortened body length, yolk edema, and spinal curvature. Notably, MP was found to induce cardiac abnormalities, including pericardial edema and decreased heart rate. Exposure to MP resulted in the accumulation of reactive oxygen species (ROS), decreased superoxide dismutase (SOD) activity, increased catalase (CAT) activity, elevated malondialdehyde (MDA) levels, and caused cardiac apoptosis in zebrafish embryos. Moreover, MP affected the transcription of cardiac development-related genes (*vmhc*, *sox9b*, *nppa*, *tnnt2*, *bmp2b*, *bmp4*) and apoptosis-related genes (*p53*, *bax*, *bcl2*). Astaxanthin could rescue MP-induced heart development defects by down-regulating oxidative stress. These findings suggest that MP induces cardiac developmental toxicity and provides additional evidence of MP toxicity to aquatic organisms.

## 1. Introduction

Organophosphorus pesticides (OPs) are widely used as insecticides in agriculture because of their low cost and effectiveness, with global annual consumption of up to 2 million tons, accounting for 38% of total global insecticide use [1,2]. The main hazard of organophosphorus insecticides is neurotoxicity, especially by blocking the phosphorylation process and inactivating acetylcholinesterase, leading to excessive accumulation of acetylcholine [3]. The extensive use of OPs chemicals has led to increased residues in crops and soil, the pollution of water, and the enrichment of the food chain via aquatic plants and animals, raising safety concerns for non-target creatures, including mankind [4,5]. According to the World Health Organization (WHO), approximately 200,000 people were poisoned annually by direct or indirect exposure to OPs [6]. Available evidence suggested that long-term exposure to OPs could cause harm to the central nervous system in people and animals, resulting in neurological illnesses such as Parkinson’s and Alzheimer’s disease, and even death [7,8]. Therefore, the toxic effects of OPs on non-target organs and other organisms should be extensively investigated.

Methyl parathion (MP) was previously one of the most widely used OPs, reaching a high of 27,563,000 pounds in US agriculture in 1971 [9]. The excessive presence of MP in the environment has led to biohazards, prompting the WHO to classify it as ecologically “extremely dangerous” in 2004 and to restrict its usage in several countries [10,11]. Exposure to MP causes neurotoxicity, developmental abnormalities, and teratogenic effects in non-target species. For example, honey bees could adsorb microencapsulated MP during foraging and carry it back to the hive, leading to a sustained slow release of the pesticide, resulting in high MP residues (4.8 mg kg^−1^) and mortality [12]. In addition, MP exposure resulted in significant reductions in body weight and growth rate, as well as an increased risk of deformity in Amarillo fish [13]. Furthermore, MP exposure was found to be genotoxic in rats, causing necrotic edema of kidney tissue and leading to severe malformations of the reproductive system [14,15]. In young children, chronic exposure to MP can result in hypersensitivity, sleepiness, memory and attention loss, and motor impairment [16]. Therefore, a comprehensive understanding of MP toxicity, especially early developmental toxicity, is needed for non-target organisms that may be exposed to MP.

MP has been detected in the aquatic environment. Previous research found that the half-lives of MP in pure water were 68 days at 8 °C (6.6 days at 25 °C), suggesting that MP residues may be persistent in cold water environments [17]. Alfonso et al. detected MP adsorption in soil samples from four locations in northern Yucatan, Mexico, with residual levels in groundwater exceeding 0.005 mg/L [18]. Arellano-Aguilar monitored MP concentrations up to 0.037 mg/L in the surface water of Salazar Dam, Mexico [13]. Methyl parathion has also been reported to accumulate in the organs of several aquatic species, such as in the tissues of *Girardinichthys multiradiatus*, a placental fish endemic to central Mexico [13], and in the kidney and digestive glands of scallops [11]. Given the accumulation of MP in the environment and the serious negative effects of MP on non-target creatures so far, a comprehensive assessment of MP toxicity to aquatic organisms is warranted.

Despite the recent development of degradation means to deal with methyl parathion contaminants, such as methyl parathion hydrolase and ozone, the untargeted toxicity of organophosphorus remains unknown [19,20]. Zebrafish, due to high fecundity, rapid embryonic development, embryonic transparency, and high experimental yield, is becoming a popular model organism in environmental toxicity studies [21,22]. For example, exposure of zebrafish embryos to 0.1 μg/L methyl parathion for 5 days resulted in increased carboxylesterase (CES) activity and downregulation of genes related to neurological function (*c-Fos*, *LINGO-1B* and *GRIN-1B*) [23], which led to a decrease in travelling distance and speed of larvae. In addition, zebrafish exposed to 5 mg/L MP for 96 h showed alterations in the expression levels of six proteins related to neural development, energy homeostasis, and cellular structure in the brain [11]. The expression of the steroidogenic acute regulatory protein (StAR) and heat shock protein 70 (hsp70) genes was also significantly reduced in the zebrafish brain after treatment with 5.2 mg/L MP [24]. However, studies on MP’s developmental toxicity in zebrafish are still limited.

In this study, we examined morphological changes in zebrafish embryonic development following MP exposure to reveal the early developmental and cardiotoxicity of MP in zebrafish. Considering that significant toxic effects may not be observed at environmental concentrations of MP in short-term exposure experiments and that the bioconcentration factor of MP was as high as 13,461 in fish [25], we used the LC50 (median lethal concentration) as the reference setting to assess the toxic effects of MP in zebrafish [26]. To investigate the potential mechanisms of MP toxicity, we further measured levels of oxidative stress and apoptosis, as well as the expression of associated genes (e.g., *nppa, bmp2b,* and *bax*). Finally, astaxanthin was shown to rescue MP-induced cardiac developmental defects by downregulating oxidative stress, which was consistent with our proposed mechanism of MP cardiotoxicity.

## 2. Materials and Methods

### 2.1. Chemicals and Reagents

Methyl parathion (CAS: 298-00-0, purity > 98%) was purchased as a solid from the Beijing North Weiye Metrology Technology Institute (Beijing, China), dissolved in pure water, and prepared as working solutions of 2.5, 5, and 10 mg/L prior to the experiments. Acridine orange (CAS: 10127-02-3) in this study was purchased from Aladdin Reagent (Shanghai, China) Co., Ltd. Astaxanthin (analytical standard, purity > 98%, CAS: 472-61-7) was purchased from Beijing Solarbio Science and Technology Co., Ltd. Kits for the evaluation of enzyme activity or other biological indicators such as CAT, SOD, ROS, MDA, Coomassie blue (for determination of total protein concentration), and Trizol reagents were purchased from Nanjing Jiancheng Bioengineering Institute (Nanjing, China). Reverse transcriptase kits and SYBR green reagents were provided by Takara (Dalian, China). All other biochemical reagents were purchased from Aladdin Reagent (Shanghai, China) Co., Ltd. and were of analytical quality.

### 2.2. Embryo Collection and Methyl Parathion Exposure

The China Zebrafish Resource Center in Wuhan provided wild-type AB strains and Tg(*myl7: EGFP*) transgenic strains of zebrafish (labeled cardiomyocytes). Prior to experiments, zebrafish were kept in recirculating water at 28 ± 1 °C for two months with a 14 h/10 h light/dark cycle and fed with Artemia salina twice daily. Female and male adult zebrafish (1:2) were transferred to a spawning box overnight and then baffles were removed the next morning to encourage spawning in the light. Zebrafish embryos developed to the 50%-epiboly phase were selected under a stereomicroscope (Olympus ZX-10, Tokyo, Japan) at 5.5–6 hpf [27]. Then, the embryos were combined and transferred to a 20 cm diameter glass dish containing the E3 solution (5 mM NaCl, 0.17 mM KCl, 0.33 mM CaCl2, and 0.33 mM MgSO4), with 200 embryos per dish, from which embryos were randomly selected for subsequent MP exposure experiment.

The treatment group received MP solutions at six concentrations (5, 10, 20, 25, 30, or 35 mg/L), whereas the control group received the E3 medium. At 5.5–6 hpf, 30 healthy embryos of each concentration of the wild-type AB strain or the Tg(*myl7: EGFP*) transgenic strain were transferred to each well of a glass 6-well plate. Three replicates per concentration were conducted. The medium was fully replenished, and deceased individuals were removed from all groups every 24 h until 96 hpf. The LC50 at 96 hpf was calculated by fitting a concentration–response curve with mortality from seven distinct concentrations following a previous method [28]. In subsequent experiments, concentrations of 2.5, 5, and 10 mg/L were chosen as equivalent to 1/10, 1/5, and 2/5 of the 96 h-LC50, respectively, following the previously reported method [29]. Thirty embryos per concentration were placed in one well of a 6-well plate and treated with 3 mL of the MP solution at concentrations of 2.5, 5, and 10 mg/L, while the control group received an equivalent volume of E3. The solutions were fully refreshed every 24 h. Three replicates per concentration were conducted. The mortality and phenotypic defects of embryos were recorded at 24, 48, and 72 hpf, and embryos were collected at 72 hpf for subsequent experimental analysis.

### 2.3. Morphological Analysis and Developmental Toxicity Assessment

Ten embryos from each replicate were randomly selected for developmental morphological analysis at 24 hpf or 72 hpf, according to established procedures [26]. At 24 hpf, the frequency of spontaneous tail coiling was counted under the microscope. At 72 hpf, the mortality, hatching rate, body length and yolk area, and development of the heart and spine were examined. The distance between the tail and the head was defined as body length and the yolk area was defined as the lateral area of the yolk. Embryos were photographed and recorded using a stereomicroscope (Olympus, SZX2), followed by quantification using ImageJ software (v1.53j, NIH, Bethesda, MD, USA) [30].

### 2.4. Assessment of Cardiac Morphology and Function

The Tg(*myl7: EGFP*) transgenic strain of zebrafish was used to assess the effects of MP on heart development and function using the previously described methods [31]. At 72 hpf, 15 embryos per replicate were randomly selected from each concentration, anesthetized with 0.0168% tricaine, and mounted on 3% methylcellulose slides. The area of the pericardium was photographed and measured using an inverted fluorescence microscope (Olympus, IX83), while ImageJ software was used to calculate each metric. The distance between the cardiac sinus venosus (SV) and the bulbus arteriosus (BA) was determined using fluorescent pictures of embryonic heart morphology taken under a fluorescent stereomicroscope with blue light stimulation. A total of 15 embryos were randomly chosen from each concentration at 48 and 72 hpf. The count of the heart rate of the embryos was evaluated for 30 s under a microscope in a constant-temperature room at 28 °C.

### 2.5. Oxidative Stress Analysis

At 72 hpf after MP exposure, oxidative stress markers such as ROS, SOD, CAT, and MDA activities in zebrafish embryos were determined following the manufacturer’s instructions [32,33]. Briefly, after exposure of zebrafish larvae to different concentrations of MP up to 72 hpf, 30 larvae of each concentration were homogenized in 150 μL of ice-cold saline and subsequently centrifuged at 4 °C at 3500 rpm for 15 min to obtain the supernatant. The total protein concentration of the supernatant was determined using a BCA protein quantification kit (Jiancheng Bioengineering Institute, Nanjing, China). To detect ROS levels, 10 embryos per replicate were chosen at random from each concentration, washed three times with the E3 medium, anesthetized with 0.04% MS-222 (buffered with sodium bicarbonate), and stained with 20 μM DCFH-DA (2′,7′-dihydrodichlorofluorescein diacetate probes) from the ROS kit for 30 min at 37 °C in the dark. Subsequently, zebrafish were mounted on 3% methylcellulose slides. Images were acquired using an inverted fluorescence microscope (Olympus, IX83), where the fluorescence intensity was quantified using ImageJ software (NIH, USA). Then, 30 embryos per replicate were selected for subsequent biochemical analysis of SOD, CAT, and MDA.

### 2.6. Acridine Orange (AO) Staining

According to a previous approach [33], acridine orange staining was used to determine apoptosis in zebrafish at 72 hpf. Five embryos per replicate were chosen from each concentration, washed three times with the E3 medium, anesthetized with 0.04% MS-222 (buffered with sodium bicarbonate), and stained with 5 mg/L AO for 30 min at 28 °C. Subsequently, zebrafish were mounted on 3% methylcellulose slides. An inverted fluorescence microscope (Olympus, IX83) equipped with a 450 nm light source was utilized to observe apoptotic cells.

### 2.7. Analysis of Gene Expression

Following the manufacturer’s procedure [32], 30 embryos per replicate were chosen at random from each concentration at the end of the exposure experiment (72 hpf) and homogenized to provide total RNA using Trizol reagent. The concentration and purity of samples were determined using NanoDrop One (Thermo Fisher Scientific, MA, Waltham, MA, United States). The RNA was diluted to 200 ng/L, then 1 μg of RNA was used for reverse transcription with the PrimeScript™ RT reagent Kit (Takara, Japan) in a 20 μL reaction volume to synthesize cDNA. The cDNA was diluted to a final volume of 50 μL. To detect gene expression, a real-time quantitative PCR method using SYBR Green dye was utilized, with β-actin serving as an internal reference. PCR with no template controls was performed in parallel for each primer pair. Three biological replicates were assayed for each treatment, and three technical replicates were run for each sample. The total volume of the q-PCR reaction system was 20 μL, including 10 μL of TB Green, 0.4 μL of PCR forward/reverse primers (10 μM), 1 μL of the cDNA template, and 8.2 μL RNase-free H_2_O (Takara, Japan). The quantitative real-time polymerase chain reaction analysis was performed in an ABI Step One plus RT-PCR system (Applied Biosystems, CA, Waltham, MA, USA) under the cycle conditions: 94 °C for 3 min, followed by 40 amplification cycles of 94 °C for 20 s, 58 °C for 30 s, and 72 °C for 20 s. The 2^−ΔΔCt^ method was applied to determine the relative expression differences of heart-related genes (*vmhc, sox9b, nppa, bmp2b, bmp4, and tnnt2*) and apoptosis-related genes (*bax, p53,* and *bcl2*). The specificity of primers was confirmed by the presence of a single peak in the melting curves of PCR reactions. Primer sequences were provided in Appendix A.

### 2.8. Rescue Experiments

Thirty well-developed embryos at 5.5–6 hpf were placed in three wells of a six-well plate, then exposed to 10 mg/L MP or a mixture of 10 mg/L MP and 30 nM astaxanthin, while the control group received the E3 medium [31]. ROS concentration, pericardial area, and apoptotic cells were photographed and examined under a stereomicroscope (Olympus, SZX2) at 72 hpf. At least 10 embryos were examined per treatment group.

### 2.9. Determination of MP Concentration in Aqueous Solution and Zebrafish Larvae

Zebrafish larvae at 72 hpf were washed three times with pure water to remove surface MP residues after exposure experiments. Forty larvae were selected for each concentration and anesthetized at 0 °C for 4 min, followed by the removal of excess water with a pipette. Larvae were homogenized at a low temperature by adding 0.4 mL of acetonitrile and then centrifuged at 14,000 rpm for 40 min. The supernatant was then transferred to a new tube, and the MP concentration in the acetonitrile solution was determined to calculate the amount in larvae. MP concentrations were determined in aqueous solutions of zebrafish embryos at 0 h and 24 h after exposure and were repeated three times.

LC-MS/MS analysis was performed using an LCMS 8050 ultra-performance liquid chromatograph–mass spectrometer with an electron spray ionization source (ESI) and a triple quadrupole mass analyzer (Shimadzu, Japan) and a TURBOVAP II nitrogen blowing concentrator (Biotage, Sweden). An ACQUITY UPLC HSS T3 (2.1 mm × 100 mm × 1.8 μm) was used as the liquid chromatographic column. Mobile phase A was water (0.05% formic acid + 2 mmol ammonium formate), and mobile phase B was acetonitrile (0.05% formic acid + 2 mmol ammonium formate) at a flow rate of 0.4 mL/min. The LC gradient used for chromatographic separation started at 100% A and decreased to 25% B in 6.5 min. From 6.5 to 9 min, the A phase decreased to 5% and was maintained for 1 min. Finally, phase A returned to 100% and was rinsed for 1 min. The following parameters were used: Electrospray source: ESI (positive ion mode); nebulizing gas: Nitrogen, with a flow rate of 3 L/min; drying gas: Nitrogen, with a flow rate of 10 L/min; collision gas: Argon, with a DL temperature of 250 °C; an interface temperature of 300 °C; a heating block temperature of 400 °C; scan mode: Multiple reaction monitoring mode (MRM), segmented acquisition. The strongest fragment was the quantitative ion, the most intense fragment was the quantitative ion, and the second strongest fragment was the qualitative ion. MP concentration detection was performed using an external standard method.

### 2.10. Statistical Analysis

Statistical analysis was conducted using the software Graph Pad Prism 8, and data were presented as mean ± standard deviation (SD). A nonlinear regression approach was utilized to provide the concentration–response curve and LC50 value. One-way ANOVA analysis was used for data with a normal distribution, followed by Dunnett’s post hoc test. The normality of data was assessed using the Shapiro-Wilk test. Bartlett’s test was used to assess homoscedasticity. The significant difference was set as * *p* < 0.05, ** *p* < 0.01, *** *p* < 0.001, or **** *p* < 0.0001.

## 3. Results

### 3.1. Stability of MP during the Experiment and Concentrations of MP in Zebrafish Larvae

In this experiment, aqueous solutions of MP at different concentrations were stable and the concentrations were nominally consistent, although the concentrations decreased slightly after 24 h (Table 1). At the end of MP, exposure to 2.5, 5, and 10 mg/L, MP concentrations in zebrafish larvae reached 0.74 ± 0.03, 2.41 ± 0.05, and 5.05 ± 0.06 μg/g, respectively.

### 3.2. MP-Induced Developmental Toxicity in Zebrafish Embryos

A concentration–response curve was established for zebrafish embryos after MP exposure at 96 hpf, and the LC50 was calculated to be 23.97 mg/L (Figure 1A). To further examine the developmental toxicity of MP, zebrafish embryos at the 6 hpf stage were exposed to different concentrations of MP (0, 2.5, 5, and 10 mg/L) up to 72 hpf, revealing that the survival and hatching rates of embryos decreased with an increasing MP concentration (Figure 1B,C). After 72 hpf exposure, the survival and hatching rates were significantly decreased in the 10 mg/L concentration group compared to the control group (*p* < 0.05 and *p* < 0.0001, respectively). Morphological abnormalities in embryos treated with 2.5 mg/L MP were not significant compared with the control group during the experiment, but MP above 5 mg/L induced developmental abnormalities in embryos, including pericardial edema, cardiac bleeding, and spinal curvature (Figure 1 D–G). The average proportions of developmental morphological abnormalities (pericardial edema, cardiac bleeding, and spinal curvature) were 35.6%, 4.4%, and 5.5% in embryos treated with 5 mg/L MP, respectively, while they increased to 84.4%, 26.7%, and 75.6% when treated with 10 mg/L (Appendix A). When compared to the control group, the body length and spontaneous tail coiling were considerably reduced in each treatment group (Figure 1H,I, *p* < 0.05, *p* < 0.01 and *p* < 0.0001, respectively). We also found a substantial increase in the yolk area of embryos in the 5 and 10 mg/L concentration groups (Figure 1J, *p* < 0.05 and *p* < 0.01, respectively), indicating a lack of yolk sac utilization.

### 3.3. MP-Induced Cardiotoxicity in Zebrafish Embryos

We investigated MP-induced alterations in zebrafish cardiac morphology using Tg (*myl7:GFP*) zebrafish transgenic lines (Figure 2A–D). Studies of the heart structure under blue excitation indicated that embryos in the low concentration group at 2.5 mg/L had normal cardiovascular curvature with a moderate amount of overlap between the ventricles and the atria, similar to the embryos in the control group (Figure 2A1,B1). However, high concentrations of MP (5 and 10 mg/L) increased the linear stretch of the heart in zebrafish, as evidenced by the absence of bending of the heart tube and the absence of distortion and elongation of the ventricles and atria (Figure 2C1,D1). After MP treatment with high concentrations of 5 and 10 mg/L, a significant increase in pericardial edema (PE) was observed when compared to the control group (Figure 2E, *p* < 0.001 and *p* < 0.0001, respectively). The distances between the embryonic cardiac venous sinus and bulbous artery bulb (SV-BA) were significantly increased following 5 and 10 mg/L exposures, as shown in Figure 2F (*p* < 0.0001 and *p* < 0.0001, respectively).

The development of the cardiac circulatory function in zebrafish was completed at 48 hpf, so we examined the heart rate in zebrafish at 48 and 72 hpf. A concentration-dependent decrease in embryonic heart rate was observed in all exposure groups. At 48 hpf, embryos exposed to 2.5, 5, or 10 mg/L MP had significantly lower heart rates than controls (Figure 2H, *p* < 0.0001, *p* < 0.0001, and *p* < 0.0001, respectively). At 72 hpf, the heartbeats of embryos from the 2.5, 5, and 10 mg/L concentration groups were significantly lower compared to controls (Figure 2I, *p* < 0.0001, *p* < 0.0001, and *p* < 0.0001, respectively). The ventricles and atria of control embryos moved rapidly and rhythmically, but those exposed to high MP moved erratically and faintly.

### 3.4. Expression of Genes Related to Heart Development

The expression of relative genes related to cardiac development was analyzed in zebrafish embryos exposed to MP at 72 hpf (Figure 3). The results indicated that the expression of *vmhc, nppa, bmp2b, vmhc,* and *sox9b* were upregulated in a concentration-dependent manner. Additionally, we observed a decrease in the expression of *bmp4* and *tnnt2*.

### 3.5. Oxidative Stress Analysis

Oxidative stress induces damage in biological DNA and cells, and its levels are often utilized to assess the toxicity of pesticides [34]. At 72 hpf, we observed a higher ROS fluorescence intensity with increasing MP exposure concentrations (Figure 4A–D). Zebrafish treated with 10 mg/L MP presented higher levels of ROS than control fish (Figure 4E, *p* < 0.0001), with considerable ROS buildup in the cardiovascular system. The SOD activity of zebrafish in the 2.5 and 5 mg/L concentration groups presented a significant increase and then experienced a significant decrease at the 10 mg/L MP concentration (Figure 4F, *p* < 0.01, *p* < 0.01, and *p* < 0.001, respectively). We also observed an increase in CAT and MDA activities with increasing concentrations (Figure 4G,H).

### 3.6. Apoptosis Analysis of Cardiac Cells

AO staining was performed on embryos at 72 hpf after MP exposure to determine whether MP could exacerbate cell damage under oxidative stress. In the control group, almost no apoptotic cells were observed (Figure 5A). However, apoptosis of cardiac cells was observed after 2.5 and 5 mg/L MP exposure (Figure 5B,C), and a substantial number of apoptotic nuclei were observed in the cardiac fraction of the 10 mg/L group of embryos (Figure 5D).

### 3.7. Expression of Apoptosis-Related Genes

To further understand the process of apoptosis triggered by MP, we examined the expression of three apoptosis-related genes. The anti-apoptotic gene *bax* was upregulated while the pro-apoptotic gene *p53* was down-regulated, according to the results of q-PCR (Figure 5E,F). At 2.5 mg/L, the anti-apoptotic gene *bcl2* was significantly down-regulated compared to the control, but strongly up-regulated at 5 and 10 mg/L (Figure 5G, *p* < 0.05, *p* < 0.01, and *p* < 0.001, respectively).

### 3.8. Astaxanthin Partially Rescued Zebrafish Embryos from MP-Induced toxicity

To verify whether ROS accumulation due to MP was one of the prime reasons for cardiac toxicity in MP exposure, we used the antioxidant astaxanthin to rescue cardiotoxicity in MP exposure. The results showed that astaxanthin could alleviate MP-induced cardiotoxicity. At 72 hpf, we observed significantly increased ROS fluorescence intensities in the MP-exposed embryos (10 mg/L MP) and astaxanthin-treated group (10 mg/L MP and 30 nM astaxanthin) compared with the control group (Figure 6A–D, *p* < 0.0001 and *p* < 0.001). Importantly, the ROS fluorescence intensity in the zebrafish heart after astaxanthin treatment was significantly lower than that of the MP-treated group (Figure 6D, *p* < 0.05). Consistent with changes in ROS, zebrafish in the MP + astaxanthin group exhibited a partial reduction in MP-induced cardiotoxicity, as evidenced by a significant reduction in pericardial area (Figure 6H, *p* < 0.0001) and a decrease in the number of apoptotic nuclei in cardiac positions compared with the MP-exposed group (Figure 6I,J).

## 4. Discussion

MP has been one of the most widely used organophosphorus pesticides in the world, and its residues and bioaccumulation in the environment pose health risks to a variety of organisms. To date, MP has been shown to induce significant neurotoxicity [23], genotoxicity [13,35], reproductive toxicity [14], and cytotoxicity [36,37] in non-target organisms, but studies on cardiotoxicity are still limited. Here, we used zebrafish to study the toxicity and mechanism of MP exposure during embryonic development. Our results demonstrated that MP induced developmental toxicity, cardiotoxicity, oxidative stress, and apoptosis in zebrafish embryos, as well as changes in associated gene expression. These findings raise concerns about the cardiogenic toxicity of MP-based organophosphorus chemicals, which should be taken into consideration when evaluating their impact on aquatic creatures.

We observed reduced embryo survival and hatching after 10 mg/L MP exposure, resulting in deformities such as shortened body length, enlarged yolk sac, pericardial edema, hemorrhage, and curved spine in zebrafish. These malformations were consistent with reports of zebrafish exposure to other organophosphorus compounds, such as monocrotophos at 36.5 mg/L and malathion at 50 mg/L [38,39]. Moreover, we evaluated the neurotoxicity of MP on embryonic development by examining the spontaneous tail-curling behavior of zebrafish embryos at 24 hpf. All concentrations of MP exposure reduced the frequency of tail curling, which was consistent with previous reports that MP was neurotoxic [23,40]. MP caused an imbalance in the body’s oxidative stress, which may lead to abnormalities in cardiac morphology and function, as well as apoptotic cell death. These findings called for a more in-depth investigation of the mechanisms of MP toxicity in the organism.

OP insecticides can lead to AChE inhibition, specifically by activating cytochrome p450 to oxidize its phosphorothioate group, generating oxygen metabolites with a strong binding affinity for acetylcholinesterase [41]. The inhibition of acetylcholinesterase activity leads to the accumulation of acetylcholine in the synaptic gap, which continuously binds to cholinergic receptors on the postsynaptic membrane, causing prolonged excitation due to the opening of sodium channels on the postsynaptic membrane, resulting in the overstimulation of neurons. However, previous studies have shown that no inhibition of AChE enzymes was observed with 100 μg/L concentrations of OPs (Diaz, dichlorvos, malathion, and methyl parathion) in zebrafish embryos during 120 h exposure experiments [23]. Another study showed that lethality rapidly exceeded the potential to induce an AChE inhibitory response with increasing concentrations of methyl parathion [41], suggesting that AChE destruction may be less pronounced due to slower metabolic processes in fish than in mammals [42]. Therefore, our study focused on identifying possible mechanisms of cardiotoxicity due to MP exposure.

The development of the zebrafish heart organ undergoes a complex sequential process regulated by multiple cardiogenic transcriptions [43], starting with a simple linear tube that curves into an S-shape, followed by cardiomyocyte proliferation for consequential organ expansion, and then to the ventricles and atria [44]. It is worth mentioning that a damaged heart normally could lead to mammalian mortality, while zebrafish with a weak or damaged heart may live for many days by obtaining adequate oxygen via passive diffusion, allowing for a comprehensive investigation of serious cardiovascular problems [45]. In our work, we discovered that MP above 5 mg/L induced pericardial edema, cardiac bleeding, the separation of atria and ventricles, and increased distance between the venous sinus and bulbous artery bulb in zebrafish embryos. In addition, at 48 and 72 hpf, all concentrations of MP induced a decrease in heart rate, suggesting that ventricular systolic function may be reduced [46]. The ventricular myosin heavy chain gene, *vmhc*, is a ventricular-specific marker in early development during the formation of atria and ventricles [47]. The relative abundance of *vmhc* was closely related to myocardial contractility [48]. *sox9b* transcriptional regulation is crucial in cardiomyocyte formation, and its aberrant expression can cause a decrease in ventricular cardiomyocytes, eventually leading to heart failure [49]. We found a significant increase in the expression of *vmhc* and *sox9b* in embryos exposed to MP up to 72 hpf, which may be responsible for the decreased contractile performance of the embryonic heart. These alterations suggested that MP significantly affected the development of zebrafish cardiomyocytes and ventricular cells.

In addition to *vmhc*, we identified aberrant expression of genes that participated in atrial and ventricular function and myocardial differentiation, such as the atrial natriuretic factor-encoding gene *nppa* and the myocyte gene *tnnt2*. *nppa* modulated cardiac hypertrophy and functions as a partial antihypertrophic factor [50,51]. In mammals, *nppa* expression was greatly reduced after birth, but it was reactivated in response to certain cardiovascular disorders [52]. On the other hand, alterations in *tnnt2* expression were intimately linked to cardiomyopathies such as hypertrophic cardiomyopathy (HCM) and dilated cardiomyopathy (DCM), both of which were significant causes of sudden cardiac death in young adults [53]. In this study, the upregulation of *nppa* expression in zebrafish embryos after MP exposure may be similar to that in mammals, and the enhancement of anti-hypertrophy ability may attenuate MP-induced cardiotoxicity. The expression of *tnnt2* was down-regulated, implying that MP exposure may result in cardiomyopathies such as HCM and DCM, and providing a plausible reason for the low survival rate in the heavily exposed group [54].

The bone morphogenetic protein (BMP) signaling pathway is essential for various stages of cardiac development, with *bmp2* required for atrioventricular canal morphogenesis and *bmp4* required for the formation of the zebrafish epicardium, vascular smooth muscle cells, and the epicardial preganglionic organ [55,56]. We observed increased expression of *bmp2b* and decreased expression of *bmp4* in exposed embryos, which correlated with pericardial edema, suggesting that MP affected important stages of cardiac development. In conclusion, MP may affect the abnormal differentiation and development of cardiac tissue by influencing the relative expression of multiple genes.

During heart development, many populations of cells are recruited by heart-related genes to the heart site, where they differentiate into various heart-related cells, including cardiomyocytes and epicardium cells [27]. Apoptotic cell death in the heart, a tightly regulated process of autonomous cell death, was commonly associated with cardiovascular diseases [57]. Abnormalities in the process of cardiac apoptosis can result in a variety of congenital cardiac abnormalities, including incorrect separation, aortic arch segmental disruption, and coronary artery anomalies. In this study, embryos exhibited apoptosis following high concentrations of MP exposure, and the apoptotic cells were largely localized in the heart region (apoptosis was observed in small amounts in the head and eye), as evidenced by AO staining. The mitochondrial pathway is the primary apoptotic pathway under oxidative stress, which includes genes that control the cell cycle and apoptosis [26,58,59]. Specifically, when cells are stimulated to start the apoptotic process, *p53* translocates from the cytoplasm to the mitochondria, suppressing the expression of anti-apoptotic gene *bcl2* and activating apoptotic gene bax [28,60]. In the present experiment, *p53* expression was downregulated and *bax* expression increased with the concentration in all exposed groups, suggesting a disruption of the mitochondrial pathway homeostasis, which may be associated with apoptosis. However, *bcl-2* expression decreased in the 2.5 mg/L group and increased in the 5 and 10 mg/L concentration groups, indicating an unusual concentration–response relationship with no trend. This anomalous concentration–response relationship was comparable to the opposing impact of low and high dosages observed by Welshons et al. suggesting a possible feedback control mechanism [61]. We speculated that MP inhibited the expression of *bcl2* at a low concentration but stimulated the expression of *bcl2* at a high concentration to prevent abnormal apoptosis, but this mechanism of action needed to be further studied.

Organophosporous compounds have been shown to decrease the activity of antioxidant enzymes and increase ROS formation [62]. Oxidative stress disorder could result in the accumulation of ROS, further leading to biological DNA and cellular damage. For example, the developmental toxicity mechanisms of two insecticides, phenoxyfen and isoxazole, in zebrafish were associated with ROS accumulation [63]. ROS levels in embryos increased significantly after high MP exposure and accumulated primarily in the heart (also observed in the head), suggesting the heart is one of the organs affected by MP [64]. Furthermore, it has been reported that ROS interfered with the normal expression of *p53* via oxidative processes, resulting in apoptosis [65]. As a result, apoptosis and the mitochondrial pathway in cardiac cells in this experiment may be triggered by high-level ROS. SOD and CAT are enzymes that scavenge excess superoxide radicals and hydrogen peroxide from cells and are important indicators of antioxidant levels [66]. In our study, MP at or above 2.5 mg/L induced a significant imbalance of oxidative stress (indicated by SOD and CAT) in zebrafish embryos. It was also shown that these two indicators related to oxidative stress were significantly altered in freshwater characid fish after 96 h in MP solutions at similar concentrations of 2 mg/L [46]. The increase in SOD and CAT activities observed in the present experiments may indicate that the organism developed a dynamic equilibrium for ROS and antioxidant defense systems in the body. The decrease in SOD activity after MP exposure at a concentration of 10 mg/L may be due to excessive depletion caused by the clearance of high concentrations of ROS in vivo. We hypothesized that as the MP concentration increased, the antioxidant system was unable to completely remove ROS from the organism, leading to a large accumulation of ROS, which, in turn, led to cardiotoxicity and even embryonic death.

To identify whether alterations in ROS levels could reduce MP-induced cardiotoxicity, we treated zebrafish with 30 nM astaxanthin. Astaxanthin is a common antioxidant with high efficiency in capturing and neutralizing free radicals [67]. The antioxidant mechanisms of astaxanthin include the direct scavenging of cellular ROS retained within and on the surface of phospholipid membranes, the protection of mitochondrial redox status and functional integrity, the enhancement of antioxidant enzyme activities such as GSH-Px and SOD [68], and the activation of antioxidant-related signaling pathways such as the Nrf-2/ARE pathway [69]. In previous studies, 30 nM astaxanthin demonstrated no cardiotoxicity in zebrafish and was able to inhibit cardiotoxicity induced by pesticides such as 1.0 mg/L benoxacor and 10 mg/L bifenadril [31,70]. In our study, astaxanthin reduced MP-induced ROS levels and alleviated pericardial edema and apoptosis. These indicators suggested that MP-induced cardiotoxicity was partially rescued when ROS levels decreased, implying that accumulated ROS contributed to MP-induced cardiotoxicity in zebrafish. We hypothesized that MP induced an imbalance of oxidative stress, leading to the accumulation of ROS, which, in turn, caused the abnormal expression of cardiac and apoptotic genes, resulting in severe cardiac developmental toxicity (Figure 7).

Regarding the enrichment of MP in the environment and organisms, future studies on the long-term exposure toxicity of MP at environmental concentrations are required to fully understand the toxicity of this compound in terms of growth, reproduction, and cross-generational inheritance, thus providing potential strategies for disease prevention caused by organisms exposed to this environmental contaminant.

## 5. Conclusions

In this study, we used zebrafish as a model to examine the developmental toxic effects of MP for the first time and revealed the potential mechanism of MP-induced cardiotoxicity. The results indicated that MP exposure could induce severe developmental toxicity, affect embryo survival and hatching, and cause effects such as shortened body length, yolk edema, and spinal curvature. In addition, MP resulted in the abnormal expression of genes, cell apoptosis, morphological malformation of the heart, and impairment of cardiac function. Importantly, MP-induced cardiotoxicity could be partially rescued by reducing the accumulation of ROS through the use of astaxanthin. Our findings provided new proof for MP’s toxicity to aquatic species, as well as new indicators for evaluating the safety of organophosphorus pesticides.

## Figures and Tables

**Figure 1 toxics-11-00084-f001:**
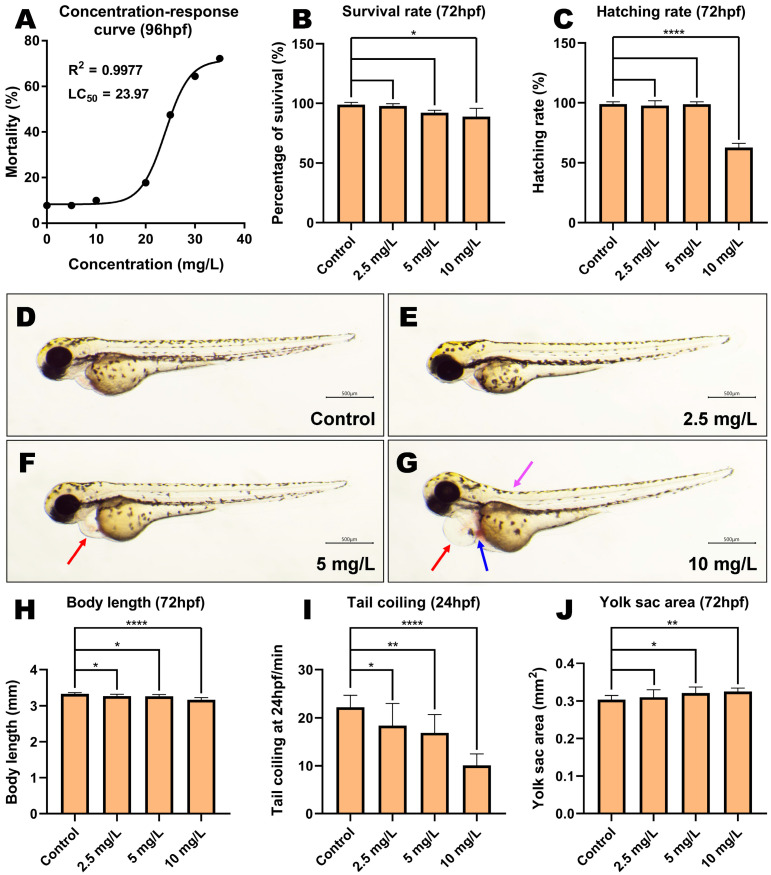
Developmental toxicity effects of methyl parathion exposure on zebrafish embryos. (**A**) Concentration–response curve of zebrafish embryos after MP exposure until 96 hpf. (**B**) Survival rate and (**C**) hatching rate at 72 hpf. (**D**–**G**) Pericardial edema was observed when zebrafish embryos were subjected to 5 and 10 mg/L MP, whereas cardiac bleeding and spinal curvature were observed when zebrafish embryos were treated with 10 mg/L. The red arrows indicate the location of the pericardial edema, the blue arrows indicate the location of the cardiac bleeding, and the purple arrows indicate the location of the spinal curvature. (**H**) Body length at 72 hpf. (**I**) Tail coiling frequency at 24 hpf. (**J**) Yolk sac area at 72 hpf. (*n* = 3, mean ± SD, ANOVA, post hoc Dunnett’s multiple comparison test, * *p* < 0.05, ** *p* < 0.01, **** *p* < 0.0001).

**Figure 2 toxics-11-00084-f002:**
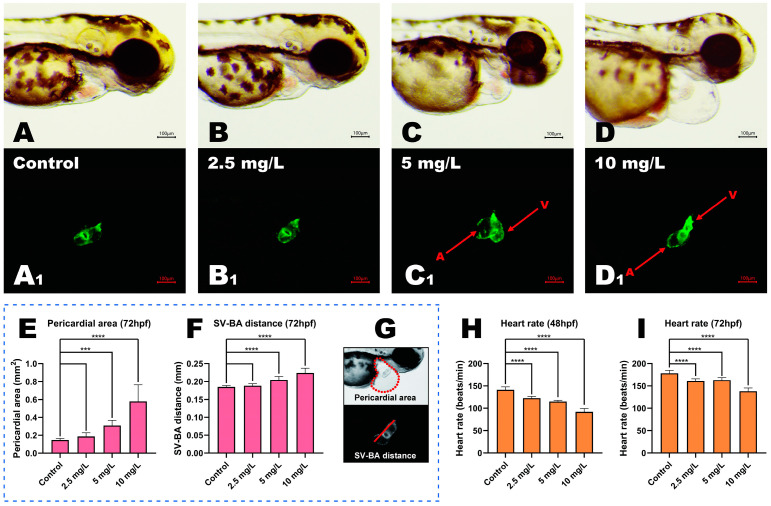
Cardiac morphological changes and functional impairment in zebrafish exposed to MP. (**A**–**D**) Brightfield microscopy of embryos at 72 hpf. (**A1**–**D1**) Fluorescent microscopy at 72 hpf; the A in the figure represents atria while V represents ventricles. (**E**) Pericardial area and (**F**) SV-BA distance at 72 hpf; SV-BA is the distance between the embryonic cardiac venous sinus and bulbous artery bulb. (**G**) Diagram showing the pericardial area and the SV-BA distance. (**H**,**I**) Heart rate at 48 and 72 hpf. (mean ± SD, *n* = 3, ANOVA, post hoc Dunnett’s multiple comparison test, *** *p* < 0.001, **** *p* < 0.0001).

**Figure 3 toxics-11-00084-f003:**
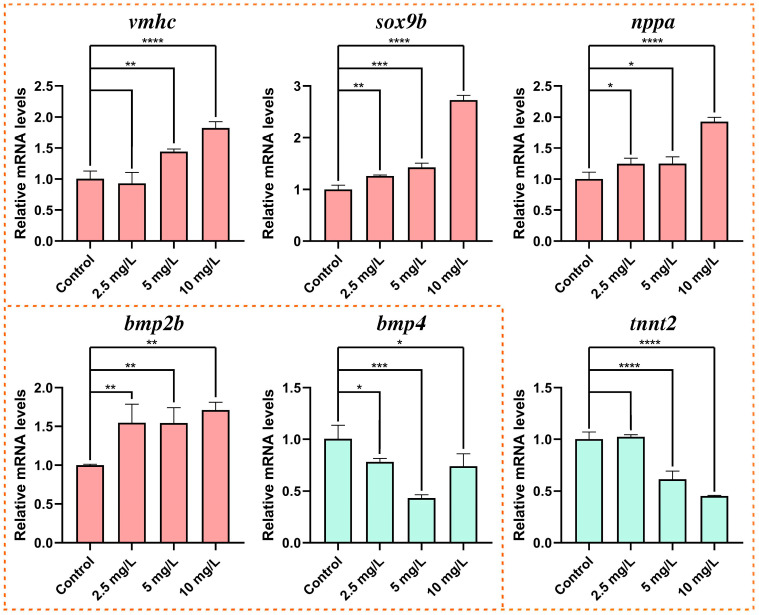
Expression of relative genes in heart development (*vmhc, sox9b, nppa, bmp2b, bmp4,* and *tnnt2*) in zebrafish embryos. (means ± SD, * *p* < 0.05, ** *p* < 0.01, *** *p* < 0.001, **** *p* < 0.0001).

**Figure 4 toxics-11-00084-f004:**
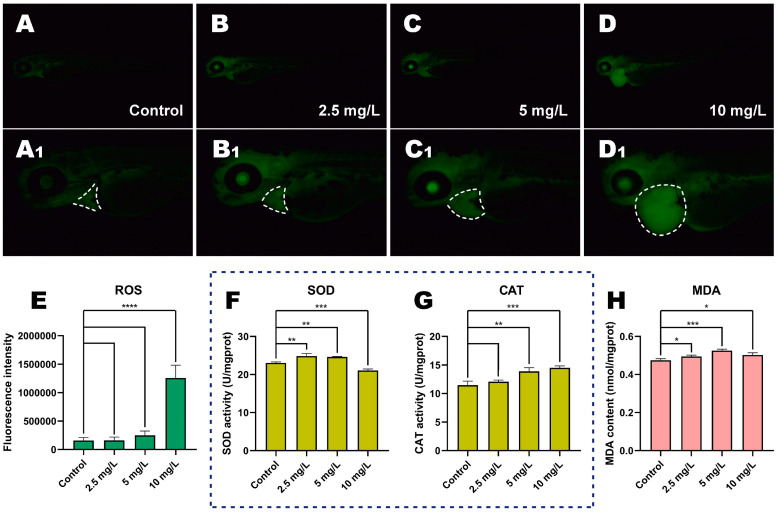
MP induces oxidative stress in zebrafish embryos. (**A**–**D**) The fluorescence distribution of ROS in embryos is presented when exposed to different concentrations of MP, where (**A1**–**D1**) indicate the magnified images of A, B, and C, respectively. The white dashed area represents the position of the heart after staining. (**E**) Relative fluorescence intensity of ROS. (**F**) Enzymatic activities of SOD. (**G**) Enzymatic activities of CAT. (**H**) Content of MDA. (mean ± SD, *n* = 3, ANOVA, post hoc Dunnett’s multiple comparison test, * *p* < 0.05, ** *p* < 0.01, *** *p* < 0.001, **** *p* < 0.0001).

**Figure 5 toxics-11-00084-f005:**
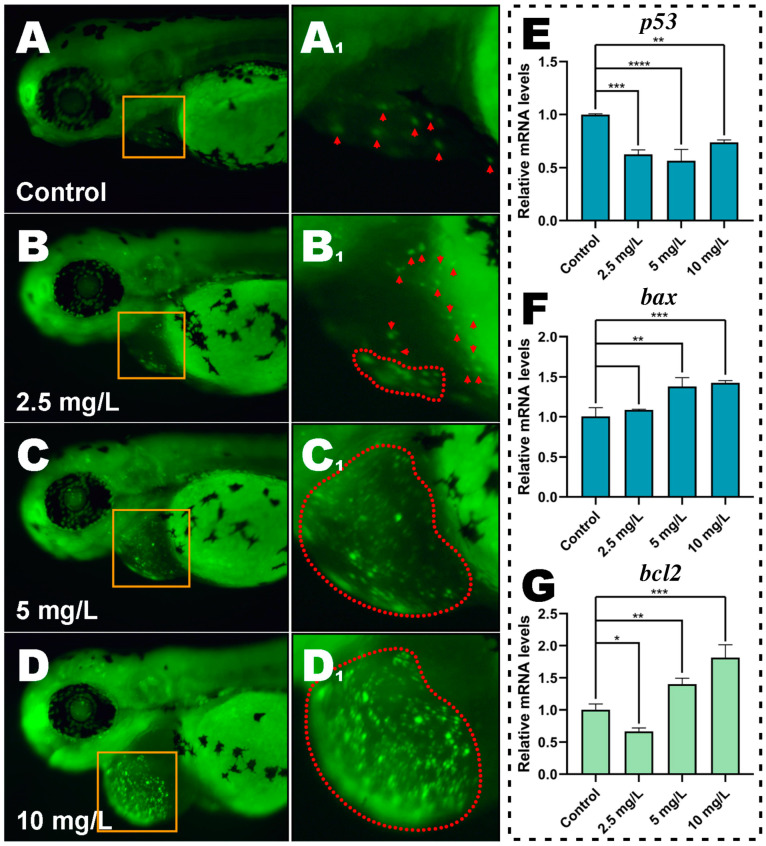
Methyl parathion induces apoptosis in zebrafish embryos at 72 hpf. (**A**–**D**) When the embryos were subjected to varied concentrations of MP after AO staining, the apoptotic cells revealed yellow-green, fluorescent patches. (**A1**) is a magnified view of the yellow area in A, and so on for the rest. The yellow-green, fluorescent highlights (i.e., apoptotic cells) that need to be attended to are also indicated in the magnified view. The red arrows point to apoptotic cells, and the red dashed area indicates a large number of dense apoptotic cells. (**E**–**G**) Expression of apoptosis-related genes (*p53, bax,* and *bcl2*) in embryos when exposed to MP at different concentrations (means ± S.D, * *p* < 0.05, ** *p* < 0.01, *** *p* < 0.001, **** *p* < 0.0001).

**Figure 6 toxics-11-00084-f006:**
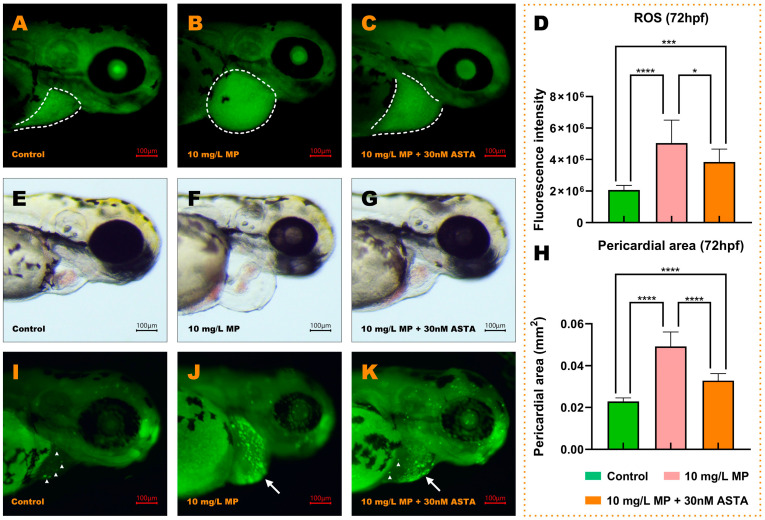
The heart morphological damage caused by MP was partially rescued by astaxanthin. (**A**–**C**) The fluorescence distribution of ROS in embryos is presented when exposed to E3, 10 mg/L MP, and the mixture of 10 mg/L MP and 30 nM astaxanthin, respectively. The white dashed area represents the position of the heart after staining. (**D**) Relative fluorescence intensity of ROS. (**E**–**G**) Brightfield microscopy of embryos at 72 hpf. (**H**) Pericardial area. (**I**–**K**) When the embryos were subjected to three concentrations after AO staining, the apoptotic cells revealed yellow-green, fluorescent patches. The short, white arrows point to apoptotic cells, and the long, white arrows indicate a large number of dense apoptotic cells. (mean ± SD, *n* = 3, ANOVA, post hoc Dunnett’s multiple comparison test, * *p* < 0.05, *** *p* < 0.001, **** *p* < 0.0001).

**Figure 7 toxics-11-00084-f007:**
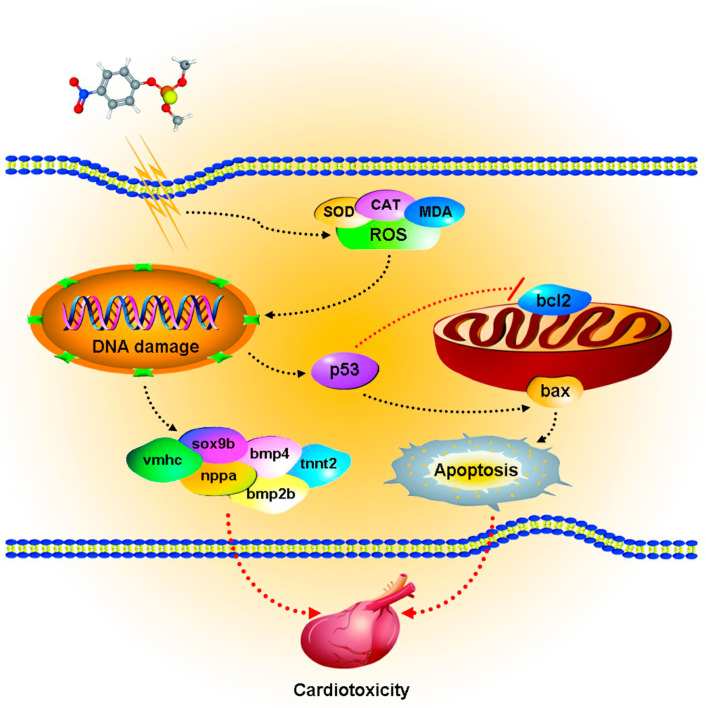
Potential mechanisms of cardiotoxicity in zebrafish exposed to methyl parathion. The apoptotic pathway induced by methyl parathion exposure in zebrafish embryonic cells disrupts the balance of oxidative and antioxidant systems, leading to ROS accumulation and DNA damage.

**Table 1 toxics-11-00084-t001:** Stability of MP during the experiment and concentration in zebrafish larvae.

Nominal Concentration (mg/L)	Time (h)	MP Concentration in Water (mg/L) ^a^	MP Concentration in Larvae at 72 hpf (μg/g) ^a^
2.5	0	2.59 ± 0.17	0.74 ± 0.03
	24	2.37 ± 0.19	
5	0	4.99 ± 0.08	2.41 ± 0.05
	24	4.84 ± 0.55	
10	0	10.70 ± 0.24	5.05 ± 0.06
	24	10.10 ± 0.32	

^a^ mean ± SD.

## Data Availability

The data presented in this study are available upon request from the corresponding author.

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
