# Peer review of "Methyl Parathion Exposure Induces Development Toxicity and Cardiotoxicity in Zebrafish Embryos"

_toxics, 2023, doi:10.3390/toxics11010084_

Round 1
Reviewer 1 Report
The authors investigated the influence of pesticide on zebrafish embryogenesis, and they focused on cardiac toxicity and apoptosis using several approaches. However, authors need to modify the manuscript. I think this manuscript acceptable after revision.
Ref 16 is lack.
l75 Girardinichthys multiradiatus should be italic.
around l85-90 It is better to describe more detail. What will happen after CES activation or genes down-regulation?
Figure 7 need to be modification. This figure provide reader for easy readability. But it looks messy. For example, the gene names cannot be seem easily because back colors are too deep. I recommend to re-make the figure simply.
I can not see supplement materials.
Reviewer 2 Report
Methyl parathion (MP) is an oragnophosphate insecticide is extremely hazardous and known to have some lethal effects on exposure.
Overall the experimental methods and conclusions are appropriate. Previously Methyl parathion (MP) exposure to zebrafish is studied. The authors concentrated this study on cardiac developmental toxicity upon exposure to MP. I would recommend authors to do gene expression analysis before and after astaxanthin exposure.
Reviewer 3 Report
The manuscript entitled “Methyl parathion exposure induces development toxicity and cardiotoxicity in zebrafish embryos” represents a remarkable contribution to the important, yet not investigated enough, topic. The issue of OPs toxicity, especially chronic exposure to low doses, is serious. This manuscript aims to investigate morphological changes in zebrafish embryonic development following methyl parathion exposure to reveal its early developmental toxicity and cardiotoxicity. To investigate the potential mechanisms of methyl parathion toxicity, the authors measured levels of oxidative stress and apoptosis, as well as the expression of associated genes (e.g., nppa, bmp2b and bax). Finally, astaxanthin was shown to rescue methyl parathion-induced cardiac developmental defects by down-regulating oxidative stress, which is consistent with the mechanism of methyl parathion cardiotoxicity proposed by the authors.
The manuscript is very well prepared. The experiments were conducted in a scientifically sound manner. The Introduction is informative and adequate. Material and Methods are given in detail. Results are presented clearly. The discussion is scientifically sound. The conclusions are supported by the results. This manuscript is a great contribution to the field. It is suitable for the Journal. Therefore, I recommend it for publication.
Round 2
Reviewer 1 Report
I have read modified version of manuscript.
Authors responded reviewer's comments appropriately.
Therefore, I think this manuscript acceptable to "Toxics".